# A Review on Experimental and Numerical Investigations of Jet Impingement Cooling Performance with Nanofluids

**DOI:** 10.3390/mi13122059

**Published:** 2022-11-24

**Authors:** Ooi Jen Wai, Prem Gunnasegaran, Hasril Hasini

**Affiliations:** 1Institute of Power Engineering, Putrajaya Campus, Universiti Tenaga Nasional, Jalan IKRAM-UNITEN, Kajang 43000, Malaysia; 2Department of Mechanical Engineering, College of Engineering, Putrajaya Campus, Universiti Tenaga Nasional, Jalan IKRAM-UNITEN, Kajang 43000, Malaysia

**Keywords:** nanofluid, jet impingement, heat transfer enhancement, swirl flow

## Abstract

Nanofluids offer great potential heat transfer enhancement and provide better thermophysical properties than conventional heat transfer fluids. Application of nanofluids in jet impingement cooling is used for many industrial and scientific purposes as it manages to effectively remove high localized heat. Owing to its tremendous improvement of the heat transfer field, the use of nanofluids in jet impingement cooling has caught the attention of many researchers. This paper reviews previous research and recent advancements of nanofluid jet impingement via both experimental and numerical studies. In experimental approaches, Al_2_O_3_-water nanofluids are the most used working fluids by researchers, and most experiments were conducted with conventional impinging jets. As for the numerical approach, the single-phase model was the preferred model over the two-phase model in obtaining numerical solutions, due to the lower computational time required. A deep insight is provided into nanofluid preparation and methods for stabilization. Parameters affecting the performance of the jet impinging system are also investigated with comparison to numerous publications. The main parameters for jet impinging include the jet-to-plate distance (H/D), the shape of the impinged plate (curved, flat or concave), nozzle configurations and the twisted tape ratio. Studies on conventional impinging jets (CIJs), as well as swirling impinging jets (SIJs), are presented in this paper.

## 1. Introduction

Rapid advancement in technology has resulted in increased demand for high performance as well as miniaturized, high-power density electronic gadgets and components. Consequently, development of thermal management techniques that fulfil all of the requirements has become extremely challenging as the cooling devices are required to remove the high heat flux that is generated. Proper heat dissipation technique is crucial as it affects the lifespan of the electronic device. It was reported that nearly 50% of electronic component malfunctions were caused by overheating or thermal failure [1]. This situation has prompted researchers from all over the world to constantly formulate new methods for development of thermal cooling. This is crucial to ensure long-term reliability of electronic devices and avoid premature failure [2].

The cooling technique adopted for most electronic devices depends on many factors such as performance, compatibility, cost, the amount of heat generation by the components, surrounding condition and maximum heat capacity. Two main cooling techniques that are widely used are air cooling (forced convection and natural) and liquid cooling (indirect and direct). For low-power consumption devices, air is often used as a coolant, either through natural or force convection. On the other hand, for high-power consumption devices, liquids are a much more suitable coolant instead, as air cooling is not sufficient in cooling a small, overheated component due to its low density. Air cooling is usually used with components with a bigger surface area and a larger mass. Because of limited air mass, confined areas are not suitable for air cooling as they can be easily heated up.

High-powered electronic devices generate high heat flux. The use of air as a cooling medium is no longer sufficient for thermal management of the system. Liquid cooling can perform better in terms of thermal management due to the higher thermal conductivity and higher specific heat capacity of liquids, which leads to higher heat transfer coefficients when compared to gas. However, there are major drawbacks and concerns with regards to the use of liquid as a coolant, such as the possibility of working fluid leakage, increased weight, corrosion, issues with electrical conductivity and pumping power required [3]. There are typically two main categories for liquid cooling techniques: indirect and direct cooling.

An indirect liquid cooling system is designed with a heat exchanger that separates the internal from the external system, thus the working fluid is not directly in contact with the electronic components. With the system separated, there is a lower risk of one system contaminating another. When compared to direct cooling, indirect cooling systems require at least one heat exchanger, thus causing loss of efficiency in the system [1]. On the other hand, direct cooling is designed with electronic components being immersed in a working fluid where heat exchange occurs. A typical working fluid in this system is dielectric liquid. This fluid has important properties such as low electrical conductivity and high thermal conductivity. A comparative study of direct and indirect cooling systems showed that direct cooling has a lower pressure drop compared to indirect cooling [2].

Many industrial applications adopt liquid cooling systems which usually consist of single-phase heat transfer fluids such as water, motor oil and ethylene glycol. Due to their relatively poor thermal conductivity, single-phase heat transfer fluids have restricted performance and are unable to meet market needs for improved performance and longer lifespans. To overcome this barrier, solid particles of nanometer size are dispersed in working fluids to increase their thermal conductivity, as solid materials naturally have higher thermal conductivities. This innovative idea was first proposed by Maxwell [3] in 1873, where a significant improvement in thermal conductivity of heat transfer fluids was observed. However, it was limited by a sedimentation problem of the solid particles. More than a century later, Masuda et al. [4] improvized the system by dispersing micrometer size solid particles instead but encountered a similar problem. The application of this innovative idea truly began when Choi [5] introduced nanoscale carbon nanotubes and metallic particles, where stability of the working fluids was significantly higher when compared to micro- or milli-sized solid particles. Researchers are drawn to investigate the untapped potential of nano-fluids because of this innovation that revolutionized heat transfer fluid.

Throughout the decades, there were several cooling methods developed to enhance thermal cooling. These include thermosyphon heat pipe that utilized latent heat of a working fluid to transfer energy through the process of evaporation. This process occurs in a boiler and condensation occurs in the condenser of the system [4]. Micro-channel heat sinks are made using high thermal conductivity materials. Coupled with state-of-the-art micro-machining technology, high surface area design for high heat flux removal can be achieved [5]. Thermoelectric cooling is another alternative that utilizes the Peltier effect. This principle is based on the change in temperature at a junction of two distinct types of semiconductors. This cooling method is not commonly used as it is expensive and inefficient, although it brings an advantage in that it does not require any moving mechanisms. Hence, this method is suitable for spacecraft heating or cooling as the absence of gravity in space makes conventional cooling with pipes impossible [6]. From the cooling techniques discussed, it is clear that numerous thermal management methods have been developed throughout the decades to effectively remove excess heat from systems. However, no single technology can reliably satisfy the ever-increasing demands for a uniform cooling process that is able to remove high heat or eliminate hot spots. Jet impingement cooling has received a lot of attention in recent years due to its capability to achieve high heat transfer rates with a variety of configurations and designs. It is also considered to be more cost-effective, requires facile installation and is lightweight compared to other cooling technologies. The working fluid for jet impingement can be phase-change material (PCM), air or even liquid coolant. The advantages and disadvantages of each cooling device is summarized in Table 1 [4,5,6,7]. Figure 1 illustrates the type of heat transfer mode against the heat transfer coefficient. From the Figure it shows that air cooling has limitations in terms of heat dissipation is not more than 100 W/cm^2^ and is therefore unable to satisfy the requirements for high heat flux devices. Jet impingement cooling is one of the top effective cooling technologies and is comparable to the other cooling methods available. 

The working fluid used in the liquid cooling system should have several properties such as high surface tension to reduce leakage, high specific heat, high thermal conductivity, high boiling point, low cost and low freezing point. Most commercial electronic devices that use liquid cooling have a closed loop system to prevent the fluid from escaping into the atmosphere. This can reduce the operating cost. In the case of an open loop system where the working fluid is lost during vaporization, a fluid making-up technique is required and thus causes the system to be more expensive and heavier [8].

Publication statistics from the field of hybrid, nanofluids and jet impingement from the years 2017–2021 is presented in Figure 2. From the author’s point of view, it shows that research concerning nanofluids is at all time high and researchers are currently exploring the hidden potential behind hybrid nanofluids, with trends increasing towards this as technology advances. While publications on nanofluids are still increasing, few works over the years have focused on the application of nanofluids in jet impingement systems. Hence, the true potential of nanofluids in jet impingement has yet to be fully discovered.

The present review paper aims to give an overview of the implementation of nanofluids in jet impingement and the effects of the geometrical parameters such as jet-to-plate distance, shape of the impinged plate (curved, flat or concave), nozzle configurations and twisted tape ratio. In this paper, the preparation methods and stability of different nanofluids and hybrid nanofluids which were prepared by different researchers have been reviewed. Additionally, the geometrical parameters of jet impingement that influence heat transfer performance, measured experimentally and numerically by various researchers, were also reviewed. Lastly, the research gaps, possible applications, challenges and future studies that revolve around the application of nanofluid jet impingement are discussed.

## 2. Introduction to Nanofluids

Due to the combination of thermal engineering and the rapid rise of nanotechnology research over the past two decades, novel heat transfer fluids known as “nanofluids” have emerged. A “nanofluid” is a heat transfer fluid that has 1–100 nm-sized “nanoparticles”, which are suspended nanoparticles, scattered throughout the base fluid. To increase the stability of the working fluid, it is crucial to make sure the nanoparticle size is smaller than 100 nm [10]. Water, oils, organic liquids (such as tri-ethylene-glycols, ethylene and refrigerants) and bio-fluids polymeric solutions are the most often utilized base fluids [11]. Numerous studies throughout the years have documented diverse nanofluid preparation methods with various nanoparticle types and their heat transfer capabilities, in addition to advancing the information about nanofluids.

### 2.1. Nanofluid Preparation Methods

Different methods of nanofluid preparation yield different thermophysical properties of nanofluids, including stability and thermal conductivity [12]. The preparation methods can also determine the particle size in the nanofluid, i.e., whether it is a micro-size or nano-size suspension, which will affect the stability of the nanofluid prepared [13]. In addition, sonication time also plays a vital role in determining the thermophysical properties of the nanofluid prepared, where an increase in the ultrasonication time and power leads to higher heat transfer enhancement, higher thermal conductivity, lower pressure drops and lower viscosity [14]. In general, there are two commonly used technique to prepare nanofluids, which are the single-step and the two-step methods. Table 2 shows the differences and a comparison between the two approaches. Regardless, both methods of preparation obtain homogeneous dispersed nanofluids that contain desirable properties and stable characteristics [15].

In the second-step method, the nanoparticles are formed followed by dispersion of the nanoparticles in the base fluid as shown in Figure 3. The single-step, or one-step, method is usually chosen for small-scale nanofluid production. It is also preferable when dealing with metallic nanoparticles, as this method manages to avoid potential particle oxidation [16]. On the other hand, the two-step method is considered more economical and it is often used for mass production. However, due to the high tendency of individual particles to agglomerate in the nanofluid before complete dispersion, it has major drawbacks. The problem worsens with increasing concentration as it directly increases the Van der Waal force of attraction, which eventually increases the agglomeration rate [17]. Hence, to reduce such occurrence in the two-step method, the nanoparticles are dispersed in the base fluid chemically or mechanically. In chemical dispersion, surfactants are added into the nanofluid to increase the stability of the suspended nanoparticles while slightly altering the viscosity and thermal conductivity of the nanofluid. As for mechanical dispersion, the nanoparticles are often dispersed through sonication. The two-step method is considered faster and simpler as it requires lesser environmental equipment [18]. Table 3 reports the summary of numerous studies on the preparation methods for the metal-based, metal oxide-based and carbon-based nanofluids conducted by researchers. 

### 2.2. Classification of Nanofluids

The classification of nanofluids is based on the type of nanoparticles chosen for nanofluid production. In general, the nanofluids can be classified into four different groups, which are (1) metal-based nanofluids, (2) metal oxide-based nanofluids, (3) carbon-based nanofluids or (4) mixed/hybrid metal-based nanofluids. The nanoparticles selected were suspended into base fluids such as oil, water or ethylene glycol. Stability of the nanofluid is crucial as it will affect the thermophysical properties if agglomeration were to occur. Hence, both physical property enhancement and stability of the nanofluid must be taken into consideration during selection of the nanofluids application. The following sections summarize each type of nanofluid described by researchers in the literature. 

#### 2.2.1. Metal-Based Nanofluids

Metal-based nanofluids are prepared by suspending metal nanoparticles such as gold, silver, aluminium, etc., in a base fluid. Beicker et al. [20] produced a gold/water nanofluid though the two-step method to study the photothermal conversion behavior of the prepared nanofluid. The investigation found that the gold nanofluid was remarkably effective even at a low concentration of 0.004% (volumetric). The prepared nanofluid was recorded to be stable for up to 120 h. 

#### 2.2.2. Metal Oxide-Based Nanofluids

The reasons behind the extensive utilization and widespread industrial applicability of metal oxide-based nanofluids are due to the high stability, suitable thermal conductivity, low cost of the nanoparticles and so forth. Due to its low cost in synthesis, it provides an economical alternative for industry. Hence, many engineering applications utilize metal oxide nanofluids as the cooling medium. Among the metal oxide nanoparticles, titania (TiO_2_) and alumina (Al_2_O_3_) are the most commonly used nanoparticles to synthesis metal oxide-based nanofluids.

#### 2.2.3. Carbon-Based Nanofluids

The majority of the articles on carbon-based nanofluids reported significant improvement in thermal –physical properties when compared to based fluid. However, the main drawback of carbon-based nanoparticles is their high cost production which limits widespread commercial use. Beicker et al. [20] studied both metal-based and carbon-based nanofluids and concluded that the MWCNT/water nanofluid would be a better economical choice compared to the gold/water nanofluid. This is because the quality of the gold/water nanofluid degrades faster and had a lower stability when compared to the MWCN/water nanofluid. Carbon-based nanofluids tend to have higher stability according to reports, as shown in Table 3. Sarafraz et al. [28] reported 504 h of stability for COOH-CNT/Water nanofluid using Nonylphenol ethoxylate (Steric) as a stabilizer, while Xie et al. [29] reported stability of up to 1440 h (2 months) for Treated CNTs nanofluid dispersed in both distilled water and ethylene glycol without adding any surfactant. However, nonpolar base fluids such as decene required a small amount of oleylamine (surfactant) for the TCNTs/decene suspensions to remain stable for months.

#### 2.2.4. Hybrid Nanofluids

Since nanofluids have consistently produced positive potential applications over time, scientists have begun to consider mixing various nanoparticles into base fluids to create what are now known as “hybrid nanofluids.” A hybrid material is something that can concurrently combine the chemical and physical properties of two or more separate materials at the molecular or nanoscale level, and it can deliver these properties in a homogenous state. The hybrid nanofluid was used primarily to obtain the properties of its constituent materials. This is because no single substance had all the necessary features to be effective for a given application. When compared to individual nanofluids, this emerging class of nanofluids shows a considerable improvement in terms of hydrodynamic properties, thermophysical properties and heat transport characteristics. Such improvement was observed in research conducted by Sun et al. [18], who applied a hybrid nanofluid containing the Ag-multiwall carbon nanotube nanoparticles at 50:50 ratio in a jet impinging cooling system. Results demonstrated that the Ag-MWCNT/water hybrid nanofluid was far more superior over the single-phased MWCNT/water nanofluid in terms of thermal conductivity. The outcomes showed that the thermal conductivity of the Ag-MWCNT/water hybrid nanofluid enhanced significantly when compared to the MWCNT/water nanofluid. Common nanofluid preparation methods such as the one- and two-step methods are used for hybrid-based nanofluid production. An overview of the general nanofluid preparation methods is given in the following section.

### 2.3. Nanofluid Stabilization Methods

It is crucial to ensure that nanofluids stability is achieved during the preparation stage to obtain optimal and equal thermophysical properties throughout the applications. A high stability nanofluid is attained when the Electrical Double Layer Repulsive Force (EDLRF) is higher than the Van der Waals force of attraction. If a higher Van der Waals force of attraction between suspended nanoparticles occurs, the agglomeration and aggregation process begins to take place, which results in clustering of the nanoparticles, which eventually leads to sedimentation over time [38]. Hence, it is crucial that the prepared nanofluid, especially if prepared through a two-step method, undergoes a stability enhancement process before it is applied to any engineering applications. Table 4 shows a summary of the nanofluid stability period, as detailed by researchers, and the properties of nanoparticles used. It can be observed that carbon-based nanofluids have the highest stability compared to other types of nanofluid. It also can be seen that preparation of metal-based nanofluids using the two-step method with appropriate surfactant can produce stable nanofluids. The following section discusses the techniques utilized in the two-step method for enhancing the stability of the nanofluid.

#### 2.3.1. Magnetic Stirrer

A magnetic stirrer which is also known as magnetic mixer is a device that is widely used in laboratories that contains a stationary electromagnet or rotating magnet. The function is to generate a rotating magnetic field, hence enhancing the homogeneity by decreasing sediment in a nanofluid. This device can be used to make a mixed solution, quickly spin, stir, immerse in a liquid or make a stir bar. Typically, the device has two knobs where the left knob is to control the stirring rate while the right knob is to control the heating rate [39,40]. This technique is often used before sonication, especially in hybrid nanofluid preparation, to mix the hybrid nanoparticles before dispersing it into the base fluid [41].

#### 2.3.2. Surfactants

The stability of a nanofluid can also be enhanced by introducing compounds known as surfactants or dispersants into the nanofluid. The presence of this compound can lower the surface tension between the nanoparticles and base fluid with the cost of deterioration of the thermophysical properties of nanofluid. This is because surfactants manage to improve the stability by preventing agglomeration and aggregation only when it is used at the optimal quantity, as excess usage may cause degradation of the chemical stability as well as decrease in thermal conductivity of the nanofluid [42,43]. The chemical properties of surfactant consist of two main parts, which are the hydrophilic polar head group followed by the long hydrocarbon chain known as hydrophobic tail. Some common surfactants used by researchers include Sodium dodecyl benzene sulfonate (SDBS), Oleic Acid (OA), Arabic gum, Polyvinylpyrrolidone (PVP), etc.

#### 2.3.3. Sonification

The process of implementing sound energy to agitate the nanoparticles is known as sonication. Nanoparticles that are subjected to sonication experience strong vibration from the ultrasonic waves which are usually higher than 20 kHz. The optimum ultrasonic time for preparation of nanofluids is yet to be fully discovered, but researchers have found that the optimum sonication time depends on the concentration and type of nanoparticles used. Higher concentration of nanoparticles often requires higher optimum sonication time. It has also been found that exceeding the optimum sonication time could decrease the stability period of nanofluid [44,45]. This method provides better dispersion when compared to magnetic stirring [41]. Two types of sonicator that are widely used by researchers to enhance the nanofluid stability are the probe type and bath type. On comparison between the two, it was reported that probe type provides better enhancement and performance when compared to bath type sonication [46].

## 3. Flow Characteristics of Jet Impingement

Figure 4 illustrates the existing jet impingement configurations that are commonly used in many studies. In general, there are five different types of jet configurations: free surface, submerged, confined, wall and plunging jet impingement. While the submerged jet happens when the working fluid is discharged into the same liquid medium, the free surface jet is generated when the liquid working fluid is ejected into ambient gas. According to the previously published literature [47,48,49,50,51], studies of potential core are more pertinent in submerged jet configurations due to the effect of the surrounding medium. As the jet nozzle exits, the entrainment of surrounding fluid into the gas medium in the free surface configuration can be regarded as insignificant. Particularly, in relation to the effects of entrainment and interaction modes, a notable contrast between free surface jets and submerged jets was observed. A closer look at the confined and unconfined submerged jet impingement systems reveals that they can be separated into two groups. In confined jet impingement, the heated working fluid is entrained and recirculated back into the impinging jet after the jet interaction with the passive ambient surroundings, resulting in the formation of recirculation zones at the outlet flow regions. However, the same concept does not occur in an unconfined jet. Thus, unconfined jets typically have higher heat transfer coefficients than confined jets. Since each has advantages of their own, both confined and unconfined jets are frequently utilized as cooling solutions. Because they are designed for compact spaces, confined impinging jets are typically used where there is little available room. As opposed to confined jets, however, unconfined jets have the benefits of easy production and simple design [47].

Conventional liquid jets often demonstrate three unique flow structures, which are the free jet, stagnation and wall jet regions, as shown in Figure 5. The working fluid that exits and diverges from the jet nozzle area is known as the free jet region. The length of this zone depends on several factors, including the shape of the jet, the distance between the jet and the target surface, and the conditions of the nozzle exit. This region can be divided into the developing flow region, fully formed flow region and potential core region. In the developing phase, the centerline velocity decreases, whereas the fully formed phase has a similar velocity profile. Additionally, the velocity in the potential core is constant and equal to the exit jet velocity. The stagnation region is created when the working fluid strikes the target surface, and it lasts until the target surface reaches zero pressure gradients. Due to the presence of an impingement wall, the flow in this area frequently exhibits a very high strain and significant curvature. In this region, the static pressure and radial velocity increases while the axial velocity sharply decreases. Finally, when the working fluid travels farther, its transverse flow velocity decreases at the wall jet zone. Due to increased turbulence intensity brought on by the sheer force exerted between the liquid and impinged wall, a higher heat transfer rate is expected at this location as reported in [52,53].

## 4. Nanofluids in Jet Impingement

In a jet impingement system, the working fluid is typically either water or ethylene glycol. The conventional working fluids could no longer meet the demands to dissipate heat as technology advanced. Thus, to replace the traditional working fluids, researchers have combined this cutting-edge working fluid with jet impingement cooling in parallel with the growth of nanofluid research. As a result of this application, the heat transfer performance is noticeably enhanced, leading to a reduction in the weight and size of the jet impingement design. This benefits the manufacturer by reducing the capital cost for manufacture of the cooling system [50]. However, such implementation of nanofluids in jet impingement systems requires proper nanofluid preparation methods before it can be practically used as a cooling agent. This is because the correct approach of nanofluid preparation is critical to ensure equal dispersion of nanoparticles in the base fluids, which will lead to higher stability. Risk of corrosion due to agglomeration may arise if the nanoparticles are not stable and cause clogging of sediment in the piping system [3]. Consequently, the heat transfer performance of the working fluids will be significantly reduced overtime. This eventually results in high cost of maintenance of the jet impingement cooling system. However, with highly stable nanofluids, they can be effectively used in jet impingement to achieve high heat transfer on the targeted surface. Many scientific and industrial applications can benefit from this, such as gas turbine cooling, high-density electrical equipment cooling or even rocket launcher cooling. 

Because of the widespread uses of nanofluid jet impinging cooling, this branch of working fluid has attracted great interest from the scientific community. Modak et al. [51] experimentally performed heat transfer enhancement in jet impingement cooling using a CuO/water nanofluid and came to three conclusions as to why the presence of nanofluid in jet impingement enhanced the heat transfer rate. First, the working fluid’s thermal conductivity is improved by the suspended nanoparticles in the base fluid. After the nanofluid exits the exit nozzle, the target surface is then bombarded with nanoparticles. This process leads to thinner boundary layers and increased turbulence and heat transfer rates. Thirdly, wettability improvement on the intended surface was cited as the cause when a thin layer of deposited nanoparticles was seen using a scanning electron microscope (SEM). In the experiment, they found that different fluids gave different contact angles, which affected the effectiveness of the cooling. Jet impingement with nanofluids gave a contact angle of 54.3° compared to 85.8° when using water impinged surface. Compared to water, nanofluid droplets spread out more widely over the surface, which increased the rate of heat transfer and is the leading potential reason behind the reduced contact angle. Further details of the experimental studies of nanofluid jet impingement cooling from past literature is discussed in the following section. 

### 4.1. Experimental Approach

In experimental analyses for nanofluid jet impingement, crucial parameters that affect the heat transfer rate are the concentrations of nanoparticles, nozzle configurations, flow rate, jet to target distance, type of base fluid, nanoparticles and size. Experimental approaches towards jet impingement can be divided into two main types, which are conventional impinging jets (CIJ) and swirling impinging jets (SIJ). The following section is divided into CIJ and SIJ.

#### 4.1.1. The Literature on Conventional Impinging Jets

Lv et al. [26] studied the jet impingement cooling with an Al_2_O_3_-water nanofluid. In the experiment, they investigated different parameters influencing the heat transfer coefficient such as different concentration, Reynolds number, jet to target distance and contact angle. Their findings showed that the presence of Al_2_O_3_ nanoparticles dispersed in water did not alter the flow characteristics, but instead enhanced the heat transfer performance. The highest heat transfer coefficient of 61.4% was obtained, and this was better than water at a jet-to-plate ratio of H/D = 4. The impact angle was low, and the highest Reynolds number and concentration were 12,000 and 2.0%, respectively. A similar experiment was conducted with a SiO_2_-water nanofluid [52]. They reported an improvement of 40% in the heat transfer coefficient compared to water coolant at the highest volume fraction (3%) with Reynolds number ranging from 8000 to 13,000. In their experiment, the jet to target distance played a vital role in heat transfer enhancement, as they observed the increasing heat transfer coefficient with increasing H/D ratio until H/D = 4, after which further increases resulted in the decrease in heat transfer coefficient. They claimed that it was because at low H/D, the jet exiting from the nozzle did not have sufficient time to fully develop, while at higher H/D, the flow may be fully developed but loses energy and thus weakens the turbulence intensity of the impinging jet. Hence, an optimum H/D is required to obtain the highest heat transfer coefficient. Barewar et al. [53] investigated the heat transfer characteristics of a free jet impinging on a copper disk plate with a ZnO nanofluid at various nanoparticle concentrations (0.02–0.1 vol%) and various H/D ratios (2–7.5) at Reynolds numbers ranging from 2192 to 9241. They achieved maximum heat transfer enhancement of 54.7% higher than pure water at stagnation zone with H/D ratio at 3.5 and volume concentration of 0.1%. A similar working fluid, a ZnO/water nanofluid, was used by Balla et al. [54] with a slot nozzle to investigate the heat transfer performance. Significant improvement in the rate of heat absorption with a staggering maximum Nu improvement of 113.9% was recorded when the nanofluid was employed in comparison to plain de-ionized water. Another experimental investigation was performed by Kareem et al. [55] on CuO/water nanofluids with different jets to target distance in a single circular jet, where it was found that an increase in H/D ratio resulted in the decrease in the Nusselt number. The data collected showed that the heat transfer rate is directly proportional to both volume fraction and Reynold number increment. They concluded the experiment with a maximum improvement in terms of Nusselt number, with 2.9% higher than pure water, and concluded that the H/D ratio played a more significant role in heat transfer enhancement than the volume fraction of the nanoparticles. The importance of volume fraction in jet impingement on a steel heated plate was further studied by Sorour et al. [56], with high concentration of SiO_2_ nanoparticles dispersed in water as base fluid. Numerous variables were analysed such as H/D ratio (0.5–8), Reynolds number (0–40,000), surface radius to jet diameter (0–8.5) and ten different concentrations (0–8.5%). In their experiment, they reported contradicting results compared to [55]; H/D was negligible in heat transfer due to small changes in heat transfer enhancement, while higher impact was seen when nanoparticles concentration was increased. The heat transfer enhancement was recorded to be 80% better than water at highest concentration of 8.5 vol %. On the other hand, an investigation into the effect of low nanoparticle concentration (0.03–0.07 wt%) was conducted by Amjadian et al. [57] on a constant heat flux aluminium disk with a Cu_2_O-water nanofluid. They found that a concentration of 0.07 wt% at Reynolds number of 7330 produced the highest heat transfer rate, and the effect of Reynolds number was more dominant compared to nanoparticle concentration. Chougule et al. [58] investigated the heat transfer characteristics of jet impingement with a CuO/water nanofluid on a hot stainless steel surface. A maximum of 75% heat transfer enhancement was observed at H/D = 4, Re = 13,000 and with highest nanoparticle concentration of 0.6%. As in the case of most jet impingement problems, they observed that highest heat transfer took place at stagnation point, and it linearly decreased as the radial distance from the stagnation point increased. 

According to numerous studies, heat transfer rate may be very high in the stagnation region under a single conventional impinging jet, but it rapidly decreases as the radial distance from the stagnation point increases. It subsequently leads to an immense gradient in the surface heat-flux across the surface where the working fluid strikes, which results in reduced performance of heat transfer. Hence, multiple impinging jets or an array of jets are proposed to control the heat flux distribution [59]. An experimental investigation was conducted to compare the heat transfer and flow region of dual synthetic jets and a single synthetic jet by Deng et al. [60]. Conclusions drawn from their investigations are that maximum Nusselt number was achieved when H/D was 5.5 for both type of jets, but higher performance was observed in the dual synthetic jets with a recorded Nusselt number of 11.4% higher compared to the single synthetic jet. Al-Zulhairy et al. studied twin jet impingement using an Al_2_O_3_-water nanofluid. Positive results were obtained from the experiment where at least 18–200% improvement could be possible when compared to pure water. This result was recorded when the parameters were H/D = 4, Re = 2000 and φ = 0.25 kg/m^3^. An array jet impingement experiment with a 36-nozzle outlet to cool photovoltaic collector, using three different nanofluids (SiC, TiO_2_ and SiO_2_) with water as a base, was investigated by Hasan et al. [61]. The SiC-water nanofluid performed better than TiO_2_/SiO_2_-water nanofluids, with lower mean temperatures recorded on the thermal collector. Another experiment which involved Cu-water nanofluids and jets array impingement cooling was conducted by Tie et al. [62]. Influence of the nanoparticles volume fraction, which ranged from 0.17 to 0.64 vol%, was studied. It was found that the dispersed nanoparticles enhanced the heat transfer except for the case of 0.17 vol%, where it was observed to have adverse effect on the heat transfer coefficient. Nayak et al. [63] conducted an experimental investigation on the dual jet configuration, where the jets were separated at a 40 mm gap from each other. Two types of nanofluids were used, Al_2_O_3_-water and TiO_2_-water. From the data collected, it was concluded that a lower H/D ratio produced a better heat transfer rate and the cooling rate decreased with an increased H/D ratio. Among the two nanofluids, Al_2_O_3_ was found to perform better and produce better heat transfer rate when compared to TiO_2_ and DI water due to better dispersion rate of Al_2_O_3_ in water. Sarkar et al. [64] investigated how different surfactants affected jet array cooling of a hot steel target plate of above 900 °C with a TiO_2_-water nanofluid as the working fluid. They found that under similar conditions, the jet array performed 60% better than a single jet using water as working fluid. They also concluded that the maximum cooling rate was achieved when Polyvinylpyrrolidone (PVP) was used as a surfactant at a flow rate of 16 l/m and H/D ratio of 15. They declared that, by using an additive based in jet array impingement, ultrafast cooling can be achieved.

Most studies on jet impingement utilize a flat target plate. However, numerous researchers have studied the effects of a conventional impinging jet on different target surfaces. Yousefi et al. [65] studied Al_2_O_3_-water nanofluid jet impingement cooling on a V-shaped plate with nanofluid volume fractions ranging from 0.02 to 0.15%. The experiment was conducted under laminar flow with five different Reynolds number values (Re = 1732, 2000, 2261, 2500 and 2719). From the experiment, it was found that at low concentration (0.02 and 0.05 vol%) and at highest Reynolds number, the highest heat transfer coefficient could be obtained. The heat transfer coefficient was 21.7% higher than that in water. Further increase in volume fraction resulted in an adverse effect to the heat transfer. It was observed that the nanofluid performed worse than water when the volume fraction was increased. Asghari et al. [66] conducted experimental work with a SiO_2_ -water nanofluid on a convex aluminium heated plate from a slot jet, with the heated plate under constant heat flux conditions. From the experimental data collected, they found that utilizing the nanofluid enhances the local and average heat transfer coefficients by 39.37% and 32.78%, respectively, compared to pure water. They also found that the heat transfer coefficient increases with the Reynolds number and concentration, which was increased from 0.1 to 1 vol%. 

Table 5 summarizes the important parameters in CIJ with a nanofluid as the working fluid, such as the jet mechanism, number and diameter of nozzle, jet-to-plate ratio (H/D) and Reynolds number. The table also includes key parameters on nanoparticles that influence the heat transfer coefficient of jet impingement such as the concentration, particle size and type of nanoparticles. The heat transfer enhancement reported from each study is also provided in the table. In short, the Reynold numbers reported from all the research papers range from 1000 to 13,000. The top three most used nanoparticles in jet impingement cooling are Al_2_O_3_, CuO and TiO_2_.

#### 4.1.2. The Literature on Swirling Impinging Jets

Twisted tape inserted into the jet nozzle acts as a swirl generator, shown in Figure 6. Usually, the tape is made from an aluminium sheet [67]. This configuration is termed as Swirling Impinging Jet (SIJ) due to the twisted tape. Another important parameter is introduced to influence the heat transfer rate, which is y/w ratio, where “y” and “w” are the width and pitch of the twisted tape, respectively. One of the earliest studies to compare the SIJs, Multi-channel impinging jets, as well as CIJs, under similar operating conditions was performed by Huang et al. [68]. The SIJs demonstrated a significant improvement in terms of Nusselt number as well as radial uniformity of heat transfer when compared to multiple channel impinging jets and CIJs.

Wongcharee et al. [23] conducted an experiment on both SIJs and CIJs with a TiO_2_-water nanofluid. Different geometrical variables such as y/w and H/D ratio, and various concentrations and Reynolds numbers were studied in their experiment to obtain the highest heat transfer enhancement. From the data collected, they found that under similar operation conditions, SIJs exhibited better overall heat transfer performance when compared to CIJs. The heat transfer performance showed an improvement with an increase in nanoparticle concentration from 0.5 to 2%. It was noted that, above that value, the performance decreases. The optimum values of y/w and H/D ratio to achieve maximum heat transfer coefficient are 6.0 and 2.0, respectively. Later that year, Wongcharee et al. [24] conducted a heat transfer enhancement study with a CuO/water nanofluid in a confined submerge type jet impingement. Three different y/w (1.43, 2.86 and 4.28) and H/D (2, 3 and 4) ratio were investigated in this study with Reynolds number ranging from 1600 to 9400 and nanoparticle concentrations of 2.0%, 3.0% and 4.0%. Under the investigated range, the optimum value that produce the highest heat transfer rate was found at y/w = 1.43, H/D = 2, Reynolds number of 9400 and nanoparticle concentration of 2.0 vol%. 

Experimental investigations on swirl flow hybrid nanofluid jet impingement were performed by Sun et al. [18]. They employed silver and multiwall carbon nanotube particles in water at seven different ratios (0:10, 1:9, 3:7, 5:5.,9:1, 7:3 and 10:0) and concentrations ranging from 0.01 to 0.05 wt%. Three types of targets surfaces were investigated which were concave, planer and concentric circular grooves. The results revealed that hybrid nanofluids enhanced the heat transfer coefficient by 29.37% and 13.35%, respectively, in SIJs when compared to deionized water. Increasing heat transfer enhancement was seen with the increase in mass fraction of nanoparticles from 0.01 to 0.05% with the maximum Nusselt number recorded at 120.53% higher than in DI water. In the case of SIJs, they tested three different y/w ratios ranging from 2 to 4, and discovered that the optimum value for y/w ratio was three. The heat transfer performance on the type of target surface varied based on the Reynolds number. The effect of the ZnO-CuO/water hybrid nanofluid in swirl flow jet impingement cooling was also investigated and a similar conclusion was obtained; that the maximum heat transfer enhancement was obtained at the configuration of H/D = 4, Re = 20,000 and concentration at 0.1% under SIJs conditions with hybrid nanofluid as the cooling agent [49]. 

Table 6 summarizes the main parameters involved in swirl flow nanofluid jet impingement studies, such as the jet configurations and properties of the nanoparticles. The results of the heat transfer enhancement are provided in the table as well. From the overall reported research, the Reynolds number ranged from 1600 to 32,000. Research into SIJs using nanofluids as a cooling agent is very limited when compared to CIJs. To date, there are only three reported studies.

### 4.2. Numerical Approach

In numerical analyses for nanofluid jet impingement, researchers often use either a single-phase model or a two-phase model to solve the complex numerical problems. For single-phase models, a few assumptions towards the nanofluids are made, such as the nanoparticles and base fluid possess the same velocity field and temperature. Thus, solid nanoparticles modelled through this approach are neglected and assumed to be one single phase with the base fluid. On the other hand, the two-phase model assumes that both the nanoparticles and the base fluid hold different temperatures as well as a different velocity field. On comparison, the two-phase model provides a more accurate estimation in terms of heat transfer coefficient and Nusselt number when compared to the single-phase model [70]. However, most researchers utilize the single-phase model due to the lower computational time and cost [71]. Table 7 summarizes all the essential parameters for both the single-phase and the two-phase model used in past studies. From the reported studies in this paper, it is shown that majority of research utilized the single-phase model approach to reduce the computational time and additionally, the numerical design is less complicated when compared to the two-phase model. 

#### 4.2.1. Single-Phase Model

A variety of equations are employed by researchers in the single-phase model to define nanofluid thermophysical properties. 

Table 8 shows the general equations used by researchers to predict the thermophysical properties of nanofluids in a single-phase model. Lorenzo et al. [72] conducted a numerical investigation with a Al_2_O_3_-water nanofluid on a flat target surface in a confined impinging slot-jet. They studied laminar flow with Reynolds numbers ranging from 100 to 400, and nanoparticle concentrations of up to 5% were investigated. It was found that the required pumping power increased by 3.9 times when compared to pure water. Reynolds number and concentration were predicted to increase. Heat transfer enhancement of 34% was recorded when H/D ratio was 10, with highest nanoparticle concentration and Reynolds number. A similar study using a Al_2_O_3_-water nanofluid with a single-phase model was conducted by Manca et al. [67]. They conducted the simulation with different nanoparticle concentrations and found that a higher pumping power was required at higher nanoparticle concentrations. At 6 vol% concentration and H/W of 10, they recorded the highest heat transfer enhancement of 18%. Manca et al. [68] also conducted a study on the performance of heat transfer using a slot jet in a confined wall. A maximum heat transfer enhancement of 36% was obtained at 5 vol% and H/W of 10. A similar problem was observed, where higher pumping power was required at higher concentrations. The influence of laminar and turbulent flow in a slot jet impinging system was numerically investigated by Dutta et al. [26]. The concentration of the alumina-water nanofluid was kept constant at 6 vol%. A maximum of 27% and 22% enhancement were achieved by laminar and turbulent flows, respectively. After conducting the performance evaluation criterion (PEC), it was concluded that nanofluids were not the best substitute for cooling fluids due to the requirement of higher pumping power. Nimmagadda et al. [68] employed magnetic fields under direct and transverse (cross flow) jet impingement conditions. Various parameters were investigated, such as type of nanoparticle (Cu, Al, TiO_2_ and Cu-TiO_2_ hybrid), Reynolds number ranging from 200 to 600, different magnetic field strengths (Ha = 0–40) and volumetric concentrations ranging from 1 to 3%. A higher average Nusselt number was shown in transverse jet when compared to direct jet impingement. The presence of the magnetic field enhanced the heat transfer process between the core fluid and interface wall. This in turn increased the flow velocity around the walls of the domain. The effect of a non-uniform magnetic field enhanced the direct jet impingement, while a transverse effect was observed in crossflow jet impingement as the magnetic field type did not inflict a higher magnetic field strength when compared to a uniform magnetic field. Maximum enhancement of 173% was obtained with 3 vol% Cu-water nanofluids under the exposure of a magnetic field. 

Numerous researchers have explored the influence of the different shape of target surface on the heat transfer rate. Ahmadi et al. [75] numerically studied the heat transfer performance using an Al_2_O_3_-water nanofluid in a semi-confined slot impinging jet on a concave shaped heated plate. Enhancement of heat transfer around the stagnation region was obtained with increasing concentration and Reynolds number but with decreasing jet-to-plate distance. An increase of 13.4% in the average Nusselt number was recorded at 5 vol% concentration and a H/D ratio of 5. The convex shaped target plate was used by Datta et al. [76] to study the behavior of a Al_2_O_3_-water nanofluid in a confined slot impinging jet. An average Nusselt number increase of 17% when compared to pure water was obtained by raising the nanoparticle volume fractions as well as the Reynolds number. Selimefendigil et al. [77] employed a heated corrugated surface to study the effect of nanoparticle shape on the thermal performance in a slot impinging jet. The nanoparticle selected in their studies was SiO_2_ with water as a base fluid. The investigated parameters included different Reynolds numbers, nanoparticle concentrations, corrugation frequencies and amplitudes, together with various nanoparticle shapes such as spherical, brick, cylindrical and brick. The highest cooling rate was achieved when cylindrical shaped SiO_2_ nanoparticle were used at the highest Reynolds number and concentration. They also found that the corrugated surface performed better in terms of heat transfer rate when compared to the flat plate. Following that, Selimefendigil et al. [78] conducted similar research but replaced the target plate with an elastic heated surface. An increase of 50.58% in the Nusselt number was observed when distinct sizes of elastic part and elastic modulus of the heated plate were used. Semi-elliptic shaped target plates were also studied by Selimefendigil et al. [79] with a CuO-water nanofluid in a slot impinging jet. A significant enhancement in the Nusselt number was obtained with a curved wall when compared to a flat wall. Twenty percent enhancement was obtained when a nanofluid was used instead of water. However, in both cases of flat and curved walls, the effect of nanoparticles showed no significant differences. 

Throughout the years, many researchers have analyzed the effects of multiple jet impingement in heat transfer performance to compare with a single jet impingement system. A study on the dual jet impingement with twisted tape was performed numerically by Amini et al. [89] on a flat heated surface. Crucial parameters that influence the heat transfer rate were investigated such as the Reynolds number, nanoparticle concentration, the jet-to-target distance and twist ratio, y/w, for swirl flow jet impingement. They found that at a higher H/D ratio of 6 and 8, higher Nusselt numbers were obtained, while at a lower H/D ratio of 2 and 4, the Nusselt numbers were lower at the stagnation region due to the presence of the twisted tape. The other parameters were observed to have the highest Nusselt number at their highest investigated value. Enhancement of 10% was achieved with dual jet when compared to single jet. The combined effect of multiple jets and four distinctive nanofluids on heat transfer over a flat copper heated plate was investigated by Kilic et al. [80]. The increase in volume fraction of the nanoparticles from 2% to 8% yielded a maximum heat transfer enhancement of 10.4% in terms of average Nusselt number. Among the tested nanoparticles, the Cu nanoparticles dispersed in water emerged as the best performing nanofluid when compared to TiO, CuO and Al_2_O_3_ nanoparticles. No significant enhancement was observed when the heat flux over the heat plate was increased by up to six times. On the other hand, the combined effect of dual jets and the nanofluid under magnetic fields was studied by Selimefendigil et al. [81]. It was concluded that the presence of a magnetic field decreased the local Nusselt number and retarded the fluid flow. However, by varying other parameters such as the jet-to-target and jet-to-jet distance, particle concentration and Reynolds number, a maximum of 46% heat transfer enhancement was achieved when compared to the base fluid. 

#### 4.2.2. Two-Phase Model

The two-phase model accounts for the interaction and slip velocity generated by Brownian effect, drag force, gravity force and other forces. Whilst in the single-phase model, those additional forces are neglected, and no slip conditions are assumed. Hence, the two-phase model requires much more computational time when compared to the single-phase model. Researchers often utilize the Eulerian–Lagrangian and Eulerian–Eulerian approaches to solve complex numerical equations under the two-phase model.

Peng et al. [82] studied jet impingement cooling by using both the single-phase and the two-phase model with an Al_2_O_3_-water nanofluid. From the data collected, the heat transfer coefficient prediction using single-phase model did not reach a satisfactory level. They concluded that to obtain more accurate data, a combination of the Eulerian–Eulerian model and the SST k-e should be deployed. They found that deploying nanoparticles in the base fluid increased the uniformity of the temperature field which eventually led to higher overall heat transfer capacity. Similarly, Abdelrehim et al. [8] also deployed both models to investigate the heat transfer of a nanofluid in a confined impinging jet. It was found that the two-phase model yielded higher local and average Nusselt numbers when compared to the single-phase model, up to a maximum enhancement of 150% at a H/D ratio of 4 and a concentration of 4%. Torshizi and Zahmatkesh [83] compared the heat transfer performance of nanofluid jet impinging using the single-phase, two-phase as well as Eulerian–Eulerian models. They concluded that the Eulerian–Eulerian model is the best approach to solve numerical problems which involved nanofluid flows, as it was more feasible to study nanoparticles in the base fluid.

The heat transfer performance and flow structure of a confined slot impinging jet was investigated by Yousefi et al. [84] using an Al_2_O_3_-water nanofluid with a two-phase model approach. An increase in the Nusselt number of 19% was recorded when the nanoparticle concentration was increased from 1% to 4% at a lower H/D ratio. It was also reported that the length of the recirculation region was reduced linearly with the increase in nanoparticle concentration. Additionally, 6% enhancement was observed when the angle of the obstacle was adjusted from 15° to 60°. An increase in Nusselt number at the stagnation point and in the average Nusselt number over the hot surface was observed. A similar heat transfer study was conducted by Huang et al. [85] in a confined circular jet. Heat transfer enhancement of up to 10% was achieved when the H/D ratio was 5, with nanoparticle volumetric concentration equal to 5%.

## 5. Conclusions and Future Direction

Jet impingement cooling is being implemented in many engineering applications due to its promising heat dissipation ability. Simultaneously, the advancement of nanofluids has caught the attention of researchers to utilize the nanofluids into the jet impinging system to enhance heat transfer performance. From the author’s perspective, the study of nanofluids’ rheological and heat transporting properties is progressing among researchers. To understand what is causing the variations in thermal conductivity, heat transfer properties and viscosity of nanofluids, more research needs to be done. It is necessary to conduct meticulous studies and analysis at the nanoscale level to reveal the intricate mechanisms and complex phenomena behind the thermophysical properties of nanofluids. Implementation of nanofluids in jet impingement cooling, causes abnormal increase in the friction factor and pressure drop. These characteristics are another major problem that must be carefully addressed as it plays a vital role in real time applications. The use of computational methods has piqued the interest of many researchers. However, more research is still required to further mitigate the drawbacks and to unravel the hidden potential of the nanofluid jet impingement cooling. This paper gives a comprehensive review on the preparation of nanofluids and its implementation in jet impingement studies, either experimentally or numerically, by researchers. The highlights of this review on nanofluid jet impingement cooling are as follows: Most researchers use a two-step method for preparation of nanofluids as it is cheaper and less complicated when compared to a single-step method;The stability of nanofluids remains one of the biggest challenges in their preparation. Although methods to enhance stability, such as surfactants and sonification are introduced, the optimum concentration of surfactant and optimum sonification time are undetermined;Limited studies have looked into the combination of hybrid nanofluids and jet impingement. Hence, efforts can be made to find out the effects of such combinations;Extensive research on the effect of impinging surfaces such as those with flat, convex or concave shape is required to further understand their jet impinging capabilities;Although application of nanofluids could enhance the heat transfer performance of jet impingement systems, agglomeration or particle sedimentation is still a challenge. Thus, more research is required to mitigate or eliminate such restrictions and to be able to deploy a better accurate prediction model for future reference;The crucial parameters that affect the heat transfer performance of a nanofluid jet impinging system include the geometrical shape of the nozzle, number of jet nozzles, ratio of jet-to-target distance (H/D), twist ratio (y/w) for SIJs and type of jet impinging system (confined, unconfined or submerged). The implementation of nanofluids in jet impingement adds a few influencing parameters affecting the heat transfer performance, such as the type of nanofluid, Reynolds number, type of flow (laminar or turbulent), nanoparticle size, shape and concentration;Few researchers have investigated nanofluids in swirling impinging jets (SIJs). The swirling effect created the tangential velocity which enhanced the heat transfer performance when compared to conventional impinging jets (CIJs). SIJs also have a higher entrainment rate and impinging area;In numerical studies, two common approaches were used to obtain a numerical solution, which are the single-phase model and the two-phase model. On comparison, most researchers utilize the single-phase model to save computational time. Computational time is higher in the two-phase model because of the introduction of additional parameters such as slip velocity and other forces, which were neglected in the single-phase model.

## Figures and Tables

**Figure 1 micromachines-13-02059-f001:**
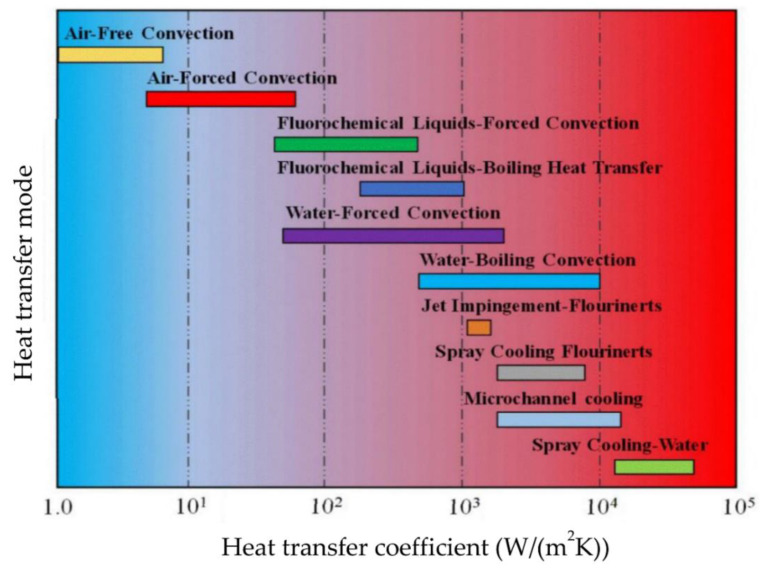
Cooling capacity comparison with various cooling methods. “Reprinted with permission from Ref [9]. Copyright 2015 Elsevier”.

**Figure 2 micromachines-13-02059-f002:**
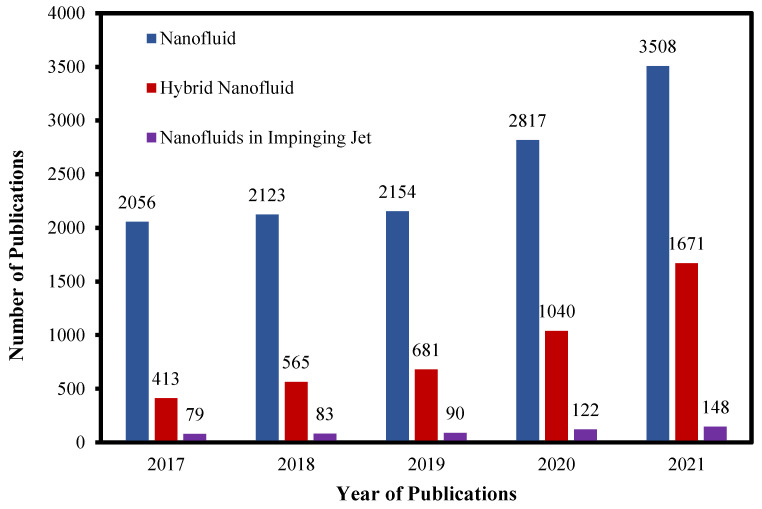
“Hybrid”, “Nanofluid” and “Jet Impingement” found in journal article titles or keywords. Searched by Science Direct 18 March 2022.

**Figure 3 micromachines-13-02059-f003:**
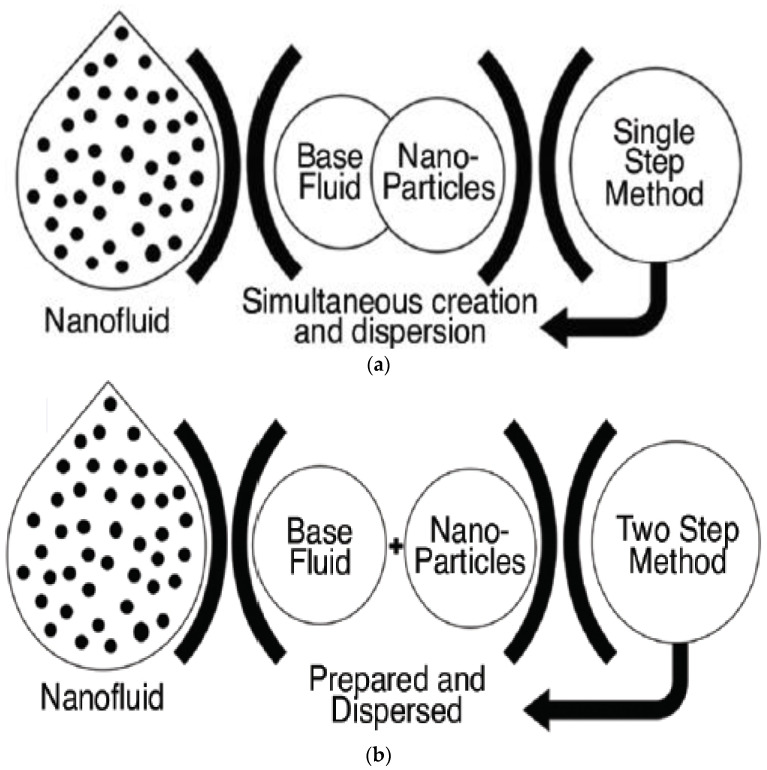
Nanofluid preparation methods (**a**) single-step method (**b**) two-step method. “Reprinted with permission from Ref [19]. Copyright 2018 MDPI”.

**Figure 4 micromachines-13-02059-f004:**
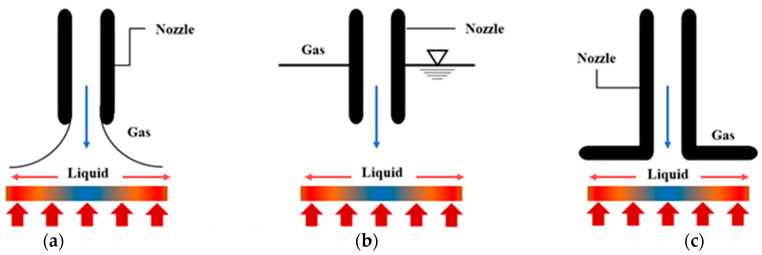
Type of jet impingement system used by researchers which consists of (**a**) Free surface jet, (**b**) Submerged jet, (**c**) Confined jet, (**d**) Plunging jet and (**e**) Wall jet.

**Figure 5 micromachines-13-02059-f005:**
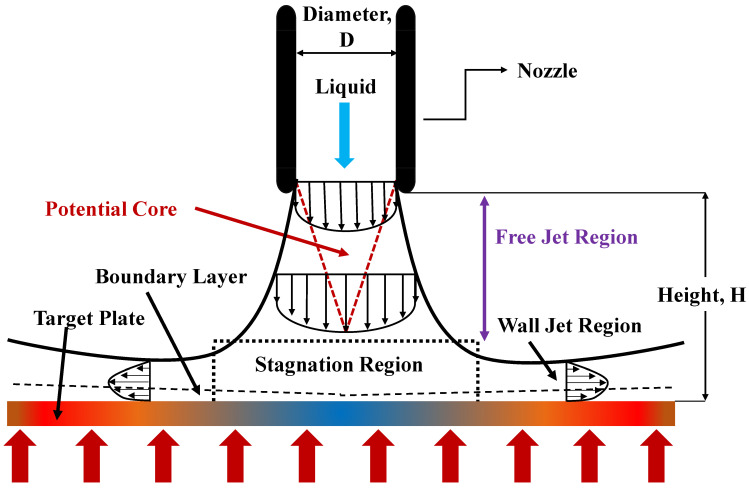
Flow structure of free surface jet. “Reprinted with permission from Ref [49]. Copyright 2022 MDPI”.

**Figure 6 micromachines-13-02059-f006:**
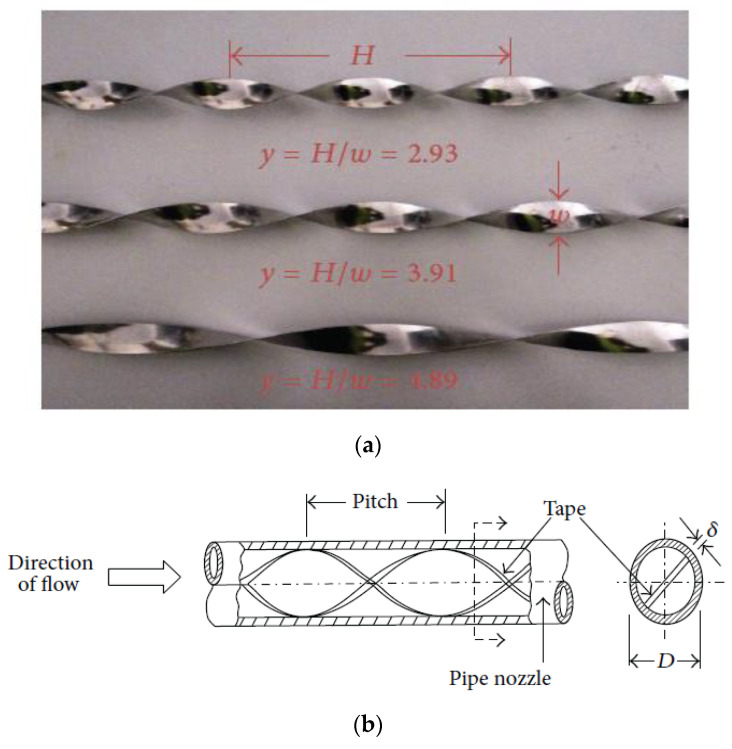
Twisted tape (**a**) at various twist ratios and (**b**) geometry drawing of twisted tape inserted into a jet impinging nozzle. “Reprinted with permission from Ref [69]. Copyright 2014, SAGE Journal”.

**Table 1 micromachines-13-02059-t001:** Summary of different cooling devices with their advantages and disadvantages in thermal management.

Cooling Device	Advantages	Disadvantages
Thermosyphon	Requires no pump for working fluid Closed loop system design reduces chance of leakages High thermal performance	It is not suitable for low temperature difference Not suitable for viscous and solid bearing fluid Liquid and vapor near critical condition cause poor performance due to their similar densities, thus lowering the driving force for recirculation
Micro-channel heat sink	Large heat exchange surface High heat transfer coefficient Small thermal gradients across the module	Coolant viscosity sets limits High streamwise temperature rise High thermal resistance High pumping power
Thermoelectric	No moving mechanism required Simple design and layout plant	Inefficient Not suitable for large area of cooling/heating
Jet impinging	TIM not required Able to target “hotspot” High average heat transfer coefficient Multiple possible configurations Multiple working fluids	Large pressure drops Not suitable for moving targets and uneven surfaces Degrade structural strength Degrade efficiency when at small spacing of jet-to-jet interaction.

**Table 2 micromachines-13-02059-t002:** Summary of differences between one-step and two-step methods.

Particular	One-Step Method	Two-Step Method
Synthesis process	Simultaneous production of nanoparticles and nanofluid	Production of nanoparticles either chemically or mechanically followed by dispersion of nanoparticles into the base fluid
Production scale	Small scale production	Large scale production
Cost of production	High cost	Low cost
Control on NPs size	Difficult and limited control over the nanoparticle size during the preparation stage	Able to control the nanoparticle size during the preparation stage
Particle oxidation	Oxidation of particles does not occur due to the elimination of drying, transportation and storage processes.	No such problem
Advantages	Reduced chances of particle agglomeration. More stable nanofluid	Cheaper and more applicable in industry
Disadvantages	Residual reactants are left in the nanofluids which might cause problems during application. Can only produce in batch	Prone to agglomeration Constant stabilizing process is needed for long term stability

**Table 3 micromachines-13-02059-t003:** Summary of preparation methods and relative surfactant used in past studies.

Nanoparticles	Base Fluid	Preparation Method	Nanofluid Sonication Time	Surfactant	Reference
Metal Based					
Au	Water	Two step	20 min	None	[20]
Au Ag	DI Water	One step	-	Cationic gemini	[21]
Cu	Methanol	Two step	30 min	APTMS	[22]
Metal-oxide Based					
TiO_2_	Water	Two step	5 h	HMDS	[23]
CuO	Water	Two step	5 h	None	[24]
Al_2_O_3_	DI Water	One step	-	None	[25]
Al_2_O_3_	Water	Two step	6 h	None	[26]
ZnO	EG	Two step	3 h	None	[27]
Carbon Based					
MWCNT	Water	Two step	20 min	None	[20]
COOH-CNT	DI Water	Two step	10 min	Nonylphenol ethoxylate	[28]
CNT	Decane	Two step	60 min	Oleylamine	[29]
MWCNT	Water	Two step	3 h	SDBS	[30]
MWCNT	Kapok seed oil	One step	6 h	none	[31]
Hybrid					
Ag-MWCNT (50:50)	DI Water	Two step	30 min	SDBS	[18]
MgO-SWCNT (80:20)	EG	Two step	6 h	None	[32]
Cu-TiO_2_ (36:64)	EG/Water (50:50)	Two step	30 min	PVP, SDBS and GA	[33]
ZnO-SWCNT (70:30)	EG/Water (40:60)	Two step	7 h	None	[34]
ZnO-MWCNT (50:50)	EG/Water (50:50)	Two Step	3 h	None	[35]
Au-TiO_2_ Au-Ag Au-Al Au-Ni	DI Water	Two Step	3 h	None	[36]
Ag-Fe_3_O_4_ (50:50)	DI Water	Two Step	3 h	None	[37]

**Table 4 micromachines-13-02059-t004:** Stability period along with nanoparticles properties from past studies.

Nanofluid	Particle Size (nm)	Concentration	Stability Period Reported	References
Metal Based				
Au/Water	10–30	0.0001–0.004 vol%	>120 h	[20]
Au/DI Water Ag/DI Water	8.6–9.4 4–33	-	80 h	[21]
Cu/Methanol	25–75	0.1–10 wt%	4320 h	[22]
Metal-oxide Based				
TiO_2_/Water	30–50	0.5–2.5 vol%	168 h	[23]
CuO/Water	30–50	2–4 vol%	168 h	[24]
Al_2_O_3_/DI Water	20	0.05–0.25 kg/m^3^	-	[25]
Al_2_O_3_/Water	30	0.5–2 vol%	480 h	[26]
ZnO/EG	10–20	1–5 vol%	6 h	[27]
Carbon Based				
MWCNT/Water	Outer D: 50–80 Inner D: 5–15 L: 10–20 (µm)	0.0001–0.03 vol%	<120 h	[20]
COOH-CNT/DI Water	D: 12–14 L: 1.5–2 (µm)	0.1–0.3 wt%	504 h	[28]
CNT/Decane	D: 15 L: 30 (µm)	0.1 vol%	1440 h	[29]
MWCNT/Water	Outer D: 50–80 Inner D: 5–15 L: 10–20 (µm)	0.1–0.5 vol%	1080 h	[30]
MWCNT/Kapok Seed Oil	D: 15.79–19.21	0.1 wt%	<720 h	[31]
Hybrid				
Ag-MWCNT/DI Water	Ag: 50 MWCNT: 20–30	0.01–0.05 wt%	48 h	[18]
MgO-SWCNT/EG	-	0.05–1 vol%	-	[32]
Cu-TiO_2_/EG-Water	Cu: 40–60 TiO_2_: <25	0.2–0.8 wt%	-	[33]
ZnO-SWCNT/EG-Water	ZnO: 10–30 SWCNT – Outer D: 1–2 Inner D: 0.8–1.6	0.05–1.6 vol%	-	[34]
ZnO-MWCNT/EG-Water	ZnO: 10–30 MWCNT - Outer D: 5–15 Inner D: 3–5	0.02–1 vol%	240 h	[35]
Au-TiO_2_/DI Water Au-Ag/DI Water Au-Al/DI Water Au-Ni/DI Water	Au: 45–85 TiO_2_: 15–40 Ag:30–65 Al: 50–75 Ni: 25–65	0.05–3 vol%	<168 h	[36]
Ag-Fe_3_O_4_/DI Water	21	0.015	-	[37]

**Table 5 micromachines-13-02059-t005:** Summary of past experimental studies on conventional impinging jet.

Jet Type	Nozzle Number	Dnozzle (mm)	H/D Ratio	Flow Rate	NanoParticles	Conc (vol%)	Diameter (nm)	Enhancement (Compared to Base Fluid)	Ref
Free	Single	1.38	6 and 8	5000 ≤ Re ≤ 12,000	Al_2_O_3_	0.15–0.6	≤50	Maximum: 62%	[51]
Free	Single	1.5	2–5	4000 < Re < 12,000	Al_2_O_3_	0.5–2	30	Maximum: 61.4%	[26]
Free	Single	1.3	2–5	8000 < Re < 13,000	SiO_2_	1–3	30	Maximum: 40%	[52]
Free	Single	0.02	2–7.5	2192 ≤ Re ≤ 9241	ZnO	0.02–0.1	25–40	Maximum: 54.7% Average: 51%	[53]
Free	Single	3X3 (slot)	2–8	5000 ≤ Re ≤ 17,500	ZnO	0.1–0.5	20	Maximum: 113.9%	[54]
Free	Single	3	13–24	1000 ≤ Re ≤ 8000	CuO	0.1–0.3	50	Maximum: 2.9%	[55]
Free	Single		-	7330 ≤ Re ≤ 11,082	Cu_2_O	0.03–0.07	15–25	Maximum: 45%	[57]
Free	Single	6	0.5–8	Re < 40,000	SiO_2_	0–8.5	8	Average: ≤80%	[56]
Free	Single	0.1	4	5000 ≤ Re ≤ 13,000	CuO	0.15 and 0.6	40	Maximum: 75%	[58]
Free	36	1	-	≤0.1666 kg/s	SiC; TiO_2_; SiO_2_	1 wt%	-	Maximum, SiC: 62.5%, Maximum, TiO_2_: 57%, Maximum, SiO_2_: 55%, Maximum, Water: 50%	[61]
Free	5	1.5	10	4000 ≤ Re ≤ 10,000	Cu	0.17–0.68	26	Maximum: 6.8%	[62]
Free	Dual	2	60–120	-	Al_2_O_3_; TiO_2_	0.01–0.07 wt%	20	Enhanced	[63]
Free	9	5	200–400	8 ≤ lpm ≤ 20	TiO_2_	56 mg/L		Maximum: 28%	[64]
Free	Single	60 (slot)	H = 10 mm; W = 1.53 mm	1700 < Re < 2800	Al_2_O_3_	0.02–0.15	15	Maximum: 21.7% Average: 13.91%	[65]
Free	Single	(slot)	H = 10 mm; W = 1.6 mm	1803 ≤ Re ≤ 2782	SiO_2_	0.1–2	7	Maximum: 39.37% Average: 32.78%	[66]

**Table 6 micromachines-13-02059-t006:** Summary of past experimental studies on swirling impinging jets.

Jet Type	Nozzle Number	Dnozzle (mm)	y/w Ratio	H/D Ratio	Flow Rate	NanoParticles	Conc (%)	Dia (nm)	Enhancement	Ref
Free	Single	8	4–7	1–4	5000 ≤ Re ≤ 20,000	TiO_2_	0.5–2.5	25 and 100	Enhanced up until a certain concentration	[23]
Confined Submerged	Single	8	1.43–4.28	2–4	1600 ≤ Re ≤ 9400	CuO	2–4	30–50	Enhanced up until a certain concentration	[24]
Free	Single	2–4	2–4	2–6	4000 ≤ Re ≤ 20,000	Ag MwCNT	0.01–0.05	Ag: 50 MWCN: 20–30	Hybrid nanofluids performed better	[18]
Free	Single	8	-	1–4	5000 ≤ Re ≤ 20,000	ZnO CuO	0.02–0.1	ZnO and CuO: <50	Hybrid nanofluids performed better	[49]

**Table 7 micromachines-13-02059-t007:** Summary of past numerical studies with single-phase and two-phase model.

Jet Type	Nozzle Number	Dnozzle (mm)	H/D Ratio	Flow Rate	NanoParticles	Conc (%)	Dia (mm)	Enhancement (Compared to Base Fluid)	Ref
Confined	Single	-	4 and 8	100 ≤ Re ≤ 400	Al_2_O_3_	1–6	30	Maximum: 32%	[72]
Confined	Single	-	4–20	5000 ≤ Re≤ 20,000	Al_2_O_3_	1–6	38	Maximum: 18%	[26]
Confined	Single	W = 6.2	4–10	100 ≤ Re ≤ 400	Al_2_O_3_	≤5	30	Maximum: 36%	[73]
Confined	Single	-	4–9.2	100 ≤ Re ≤ 30,000	Al_2_O_3_	3–6	47	Laminar Flow: 27% Turbulent Flow: 22%	[74]
Semi-confined	Single	W = 5	5 and 10	4740 and 9000	Al_2_O_3_	≤5	36	Stagnation: 11.2% Average: 13.4%	[75]
Confined	Single	-	3 and 7	5000 ≤ Re ≤ 20,000	Al_2_O_3_	≤6	15–25	Maximum: 17%	[76]
Confined	Single	-	5	100 ≤ Re ≤ 500	SiO_2_	≤4	-	Maximum: 43.07% Average: 14.28%	[77]
Confined	Single		10	25 ≤ Re ≤ 200	SiO_2_	1 and 4	-	Average: 30.65%	[78]
Confined	Single	-	4	100 ≤ Re ≤ 500	CuO	≤5	29	Average: 20%	[79]
Confined	Multiple	-	-	V < 11 m/s	Cu, CuO, Al_2_O_3_, TiO	2–8	25	Cu compared to water: 9.6% Cu compared to CuO: 2.2% Cu compared to Al_2_O_3_: 4.6% Cu compared to TiO: 5.1%	[80]
Confined	Multiple		4–16	100 ≤ Re ≤ 400	Al_2_O_3_	≤4	-	Average: 46%	[81]
Semi-confined	Single	-	-	2000 ≤ Re ≤ 16,000	Cu	1.5–3	25	Maximum: 30%	[82]
Confined	Single	-	0.5–4	100 ≤ Re ≤ 400	Al_2_O_3_	1–4	20	Maximum: 150%	[8]
Confined	Single	W = 6.2	4	100 ≤ Re ≤ 500	Al_2_O_3_	≤5	30	Enhanced	[83]
Confined	Single	W = 10	1–10	5000 ≤ Re ≤ 15,000	Al_2_O_3_	1–4	20	Average: 6%	[84]
Confined	Single	2	1–5	5000 ≤ Re ≤ 30,000	Al_2_O_3_	1–5	-	Average: 15%	[85]

**Table 8 micromachines-13-02059-t008:** Models applied for thermophysical properties of nanofluids and hybrid nanofluids [11,86,87,88].

Property	Nanofluid	Hybrid Nanofluid
Density (ρ ρ)	$\left(1-\phi_{1}\right)\rho_{f}+\phi_{1}\rho_{s1}$	$\left[\left(1-\phi_{2}\right)\left\{\left(1-\phi_{1}\right)\rho_{f}+\phi_{1}\rho_{s1}\right\}\right]+\phi_{2}\rho_{s2}$
Viscosity (µ)	$\frac{{µ}_{f}}{{\left(1-\phi_{1}\right)}^{2.5}}$	$\frac{{µ}_{f}}{{\left(1-\phi_{1}\right)}^{2.5}{\left(1-\phi_{2}\right)}^{2.5}}$
Heat capacity (ρ C_p_)	${\left(1-\phi \rho C_{p}\right)}_{f}+\phi_{1}{\left(\rho C_{p}\right)}_{s1}$	$\left[\left(1-\phi_{2}\right)\left\{\;{\left(1-\phi \rho C_{p}\right)}_{f}+\phi_{1}{\left(\rho C_{p}\right)}_{s1}\right\}\right]+\phi_{2}{\left(\rho C_{p}\right)}_{s2}$
Thermal conductivity (*k*)	$\frac{k_{s1}\;+\;2k_{f}\;-\;2\phi_{1}\left(k_{f}\;-\;k_{s1}\right)}{k_{s1}\;+\;2k_{f}\;+\;\phi_{1}\left(k_{f}\;-\;k_{s1}\right)}\;\times \;k_{f}$	$\frac{k_{s2}\;+\;2k_{nf}\;-\;2\phi_{2}\left(k_{nf}\;-\;k_{s2}\right)}{k_{s2}\;+\;2k_{nf}\;+\;\phi_{2}\left(k_{nf}\;-\;k_{s2}\right)}\;\times \;\frac{k_{s1}\;+\;2k_{f}\;-\;2\phi_{1}\left(k_{f}\;-\;k_{s1}\right)}{k_{s1}\;+\;2k_{f}\;+\;\phi_{1}\left(k_{f}\;-\;k_{s1}\right)}\;\times \;k_{f}$

## Data Availability

Data are available on request due to restrictions. The data presented in this study are available on request from the corresponding author. The data are not publicly available due to privacy and ethical concerns.

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
