# Peer review of "A Review on Experimental and Numerical Investigations of Jet Impingement Cooling Performance with Nanofluids"

_micromachines, 2022, doi:10.3390/mi13122059_

Round 1
Reviewer 1 Report
The paper presents a review on the preparation of nanofluid with it is implementation in jet impingement study. The paper is well structured. It needs an extensive language editing and proof reading. The main concern the reviewer has is that the authors do not provide any research outlook in their review. Also, connecting this review with application would be very beneficial.
Other than that, here are some minor modifications:
- In table one, make enough space between the columns as the text is currently too close to each other.
- In figure 5, specify the numbering a,b,c,d,e on the figure.
- In figure 6, add “D” after diameter
- In table 5, reference 75, I guess you mean 27
- In table 5, the enhancement is compare to what? (water cooling?)
- In table 6, check the reference numbers
- Again, the paper needs and extensive proof reading and language editing.
Author Response
The authors are highly obliged and thankful to you for the constructive and valuable comments to improve the quality of the paper. The manuscript has been revised accordingly and the actions taken are listed in the attachment. Kindly refer the attachment.
Thank you.

Reviewer 2 Report
This paper reviewed the progress made towards the jet impingement cooling system using nanofluids through experimental and numerical approaches. The article focused on reviewing the preparation of nanofluids, the main influencing parameters that affect the jet impingement cooling performance, and different types of impinging jets. The reviewer would like to pass the following comments to the authors. When these issues have been addressed, this manuscript may be accepted.
1. There are many typographical and grammatical errors in the article, and some of them affect the reading experience. In addition, when presenting past studies, the usage of past and present tense is mixed. The reviewer suggests the authors carefully check and proofread the manuscript and fix these issues.
2. Large portion of the manuscript has been dedicated into reviewing the preparation of the nanofluids, however, it is less clear that how differently prepared nanofluids would affect the jet impingement cooling performance. It would be helpful to discuss more.
2. Tables 1 and 2 contain good information, but they could be more concise.
3. Does Figure 4 include any new information compared to Table 3? It seems to be redundant. If the information is repeated, it may be removed.
4. More recent/latest studies should be included in the review.
5. In the Conclusion and future directions, the reviewer found the summary of the review, however, the discussion on the future research direction of the jet impingement cooling is insufficient. The reviewer suggests the authors to add relevant discussion.
Author Response
The authors are highly obliged and thankful to you for the constructive and valuable comments to improve the quality of the paper. The manuscript has been revised accordingly and the actions taken are listed in the attachment. Kindly refer to the attachment.
Thank you.

Reviewer 3 Report
Jet impingement cooling and nanofluids are a very important topics. The article submitted for review is interesting and prepared quite carefully. I have minor comments:
- please add missing explanations of the symbols used (e.g. H / D, ...)
- please edit the section: Conclusion and future directions (text alignment problem)
- please correct references according with the guidelines of journal.
In my opinion, the article is suitable for publication in the presented form.
Author Response

(The authors gave the same response as above.)

Round 2
Reviewer 1 Report
The Authors responds to the reviewer's minor comments, however, they have not response to the major requested modifications mentioned in the first report:
1- It needs an extensive language editing and proof reading.
2- The main concern the reviewer has is that the authors do not provide any research outlook in their review.
3- Also, connecting this review with application would be very beneficial"
Please revise the manuscript accordingly
Author Response
Dear Sir/Madam,
Many thanks for your valuable comments and suggestions, We have thoroughly revised our manuscript according to your suggestions. The actions taken are listed in the attachment. Kindly refer the attachment.

Reviewer 2 Report
I appreciate the authors' response to my comments and I think the comments have been addressed. Therefore, I would like to recommend the acceptance of the manuscript.
Author Response
Many thanks for your valuable comments and suggestions.
Round 3
Reviewer 1 Report
Thank you for your effort.
Even the reviewer like the structure of the paper, the authors response to the reviewer's main concerns was not sufficient.
This review does not provide any real research outlook or new perspectives. The review does not discuss seriously any connection with applications to highlight the shortage in the knowledge.
These points make the reviewer questioning the adding value of this review.